# Generation and Consolidation of Recollections for Efficient Deep Lifelong Learning

## Abstract

Deep lifelong learning systems need to efficiently manage resources to scale to large numbers of experiences and non-stationary goals. In this paper, we explore the relationship between lossy compression and the resource constrained lifelong learning problem of knowledge transfer. We demonstrate that lossy episodic experience storage can enable efficient knowledge transfer between different architectures and algorithms at a fraction of the storage cost of lossless storage. This is achieved by introducing a generative knowledge distillation strategy that does not store any training examples. As an important extension of this idea, we show that lossy recollections stabilize deep networks much better than lossless sampling in resource constrained settings of lifelong learning while avoiding catastrophic forgetting. For this setting, we propose a novel dual purpose recollection buffer used to both stabilize the recollection generator itself and an accompanying reasoning model.

## 1 Introduction

In this work, we focus on developing a biologically motivated deep neural network architecture to accomplish the task of general purpose knowledge transfer. The goal is to build a network that can not only learn a skill, but also efficiently transfer the function it learns to other networks with a wide variety of a different possible architectures. A neural network with this ability is able to readily apply prior knowledge while optimizing its own learning architecture for the current task of interest. One useful application of this ability would be for a lifelong learning system to gradually modify its architecture for a task to be more efficient in terms of storage and compute over time. Humans seem to do this. They use less and less mental capacity over time as a skill is mastered. Another important application of this ability is to constrain the learning process for a neural network to allow rapid learning without catastrophic forgetting (McCloskey & Cohen, 1989). A popular theory of human cognition proposed by McClelland et al. (1995) hypothesizes that a similar knowledge transfer process between the hippocampus and neocortex may account for the lack of catastrophic forgetting in humans. In both of these applications, current successful general purpose deep learning techniques are highly reliant on direct storage of full experiences. Generally, knowledge distillation techniques (Bucilu et al., 2006; Hinton et al., 2015) are focused on using large quantities of real unlabelled or labelled in domain data to transfer a function across neural networks. Additionally, top techniques for preventing catastrophic forgetting with application to supervised continual lifelong learning (Rebuffi et al., 2017; Lopez-Paz & Ranzato, 2017) and reinforcement learning (Lin, 1992; Mnih et al., 2015) rely on explicit storage for replay of full past experiences. In this paper, we consider how to use modern deep neural networks to create an efficient and scalable form of lifelong episodic experience storage with these applications in mind.

The major contribution of this work is the development of a paradigm for storage of lossy experiences (i.e. recollections) using variational autoencoders (Kingma & Welling, 2014) in a continual lifelong learning setting. We demonstrate that recently proposed discrete latent variable variational autoencoder models (Jang et al., 2017; Maddison et al., 2017) are capable of significantly improved sample compression at the same reconstruction distortion as their very popular continuous variable counterparts. In this work, we also highlight an inherent trade-off between capacity and exploration when using VAEs for knowledge transfer. Our proposed recollection buffer is critical for bypassing this trade-off. Ultimately, our lossy experience storage is able to store a larger diversity of examples at a given resource constraint and consistently perform better than lossless storage in preventing

catastrophic forgetting for resource limited many task lifelong learning problem settings. Even more encouraging is that lossy storage is very scalable. We find that the marginal footprint of lossy storage decreases as the number of tasks and experiences grow.

## 2 ACHIEVING SPACE EFFICIENT EPISODIC STORAGE

Recent work on continual lifelong learning in deep neural networks (Rebuffi et al., 2017), (Lopez-Paz & Ranzato, 2017) has focused on the resource constrained lifelong learning problem and how to promote stable learning with a relatively small diversity of prior experiences stored in memory. In this work, we complete the picture by also considering the relationship to the fidelity of the prior experiences stored memory. We achieve this by considering the resource constraint in scaling lifelong learning not in terms of the number of full examples stored, but instead in terms of bits of storage. This is very practical because it relates back to real footprints on computer hardware. We would like to scale to an unbounded number of examples, but in practice we must ground our work in finite time horizons. With this in mind, what is important is to demonstrate that even if the startup cost is high, the incremental maintenance cost gets lower as the number of experiences grows.

### 2.1 EXPERIMENTAL SETTINGS

We consider two types of evaluation settings to separately test the capabilities of our model for *knowledge distillation* and *continual lifelong learning*. For knowledge distillation we evaluate our ability to use a generative model and teacher supervised model together to produce good input and output pairs for efficiently communicating knowledge to a student neural network. In all of our knowledge distillation experiments, we report an average result over 5 runs. For continual life-long learning we evaluate our ability to stabilize the training of a neural network over a sequence of many tasks by interleaving recollections of past experiences and associated labels with incoming examples. We first calculate the maximum allowable experience buffer size for a given resource constrained architecture, and then use reservoir sampling to maintain the buffer once we reach capacity. More details can be found for all of our experiments in Appendix A.

We will empirically support the methods of this paper by applying them to the popular MNIST digit recognition dataset (Lecun et al., 1998) as well as a multi-task split of the CIFAR-100 image recognition dataset (Krizhevsky, 2009) considering each of the 20 course grained labels to be a task, and the Omniglot character recognition dataset (Lake et al., 2011) considering each of the 50 alphabets to be a task. For MNIST and Omniglot we follow prior work and consider 28x28 images with 1 channel and 8-bits per pixel. [1] CIFAR-10 is leveraged to test the efficacy of transfer learning on CIFAR-100 as the dataset represents images of the same structure drawn from a disjoint set of labels.

We consider two types of resource constraints in our experiments. An *incremental storage constraint* views the initial size of the system as a sunk cost and isolates the effect of incremental scaling with an increasing number of experiences. In contrast, a *total storage constraint* considers all bits of storage used as part of the constraint. It is informative to see how lifelong learning models perform in both of these settings for a given finite learning interval. On the one hand, incremental storage constraints can be misleading, especially for short time horizons, and favor models that are not truly apples to apples in terms of storage used. On the other hand, the lifelong learning setting we strive for has a larger number of experiences than those of all previous experiments in literature. In this extreme setting, incremental behavior will have a dominant effect on performance.

### 2.2 COMPRESSING RECOLLECTION STORAGE WITH DISCRETE LATENT VARIABLES

Typical experience replay strategies that, for example, store 32x32 CIFAR images with 3 color channels and 8-bits per pixel per channel will incur a cost per image stored of 8x3x32x32 = 24,576 bits. Deep non-linear autoencoders are considered a non-linear generalization of PCA and are a natural choice for compression problems. Theoretically, an autoencoder with a representation the same size as its input should be able to copy any input by simply learning an identity transformation.

---

[1] MNIST and Omniglot images were originally larger, but others have found the down sampling to 28x28 does not effect performance of models using it to learn.

An autoencoder with a continuous latent variable of size $h$, assuming standard 32-bit representations used in modern GPU hardware, will have a storage cost of $32h$ bits for each latent representation.

Unfortunately, continuous variable autoencoders which use 32-bits for their network parameters may incur an unnecessary storage cost for many problems. We propose a principled approach to guarding against this issue by leveraging the recently proposed variational autoencoder model with categorical latent variables (Jang et al., 2017; Maddison et al., 2017) to enable the model to learn a representation while explicitly considering a certain degree of storage precision. For a categorical latent variable autoencoder, we consider a bottleneck representation between the encoder and decoder with $c$ categorical latent variables each containing $l$ dimensions representing a *one hot* encoding of the categorical variable. By varying the size of $l$ we can explicitly control the precision of the representation to match the requirements of the problem and keep storage costs low. This is because while this representation requires $S_{1h} = lc$ bits to store the concatenated latent variables, with simple binary encoding of each categorical latent variable, we can store this representation with the following number of bits (Cover & Thomas, 2012):

$$S_{\text{sbe}} = c \cdot \lceil \log_2(l) \rceil . \tag{1}$$

We can think of every autoencoder as having two major components called the encoder and decoder, which should have specialized architectures tuned to the problem of interest. In the standard formulation, the representation learned by the encoder $z$ is generally modeled with continuous variables. In order to model an autoencoder with discrete latent variables, we follow the success of recent work (Jang et al., 2017; Maddison et al., 2017) and leverage the Gumbel-Softmax function. The Gumbel-Softmax function leverages the Gumbel-Max trick (Gumbel, 1954; Maddison et al., 2014) which provides an efficient way to draw samples $z$ from a categorical distribution with class probabilities $p_i$ representing the output of the encoder:

$$z = \text{one\_hot}(\underset{i}{\text{argmax}}[g_i + \log(p_i)]) \tag{2}$$

In equation 2, $g_i,...,g_d$ are samples drawn from Gumbel(0,1), which is calculated by drawing $u_i$ from Uniform(0,1) and computing $g_i$=-log(-log($u_i$)). The *one\_hot* function quantizes its input into a one hot vector. The softmax function is used as a differentiable approximation to argmax, and we generate $d$-dimensional sample vectors $y$ with temperature $\tau$ in which:

$$y_i = \frac{\exp((g_i + \log(p_i))/\tau)}{\sum_{j=1}^{d} \exp((g_j + \log(p_j))/\tau)} \tag{3}$$

The Gumbel-Softmax distribution is smooth for $\tau > 0$, and therefore has a well-defined gradient with respect to the parameters $p$. During forward propagation of our categorical autoencoder, we send the output of the encoder through the sampling procedure of equation 2 to create a categorical variable. However, during backpropagation we replace non-differentiable categorical samples with a differentiable approximation during training as the Gumbel-Softmax estimator in equation 3. Although past work (Jang et al., 2017; Maddison et al., 2017) has found value in varying $\tau$ over training, we still were able to get strong results keeping $\tau$ fixed at 1.0 across our experiments. Across all of our experiments, our generator model includes three convolutional layers in the encoder and three deconvolutional layers in the decoder.

In left chart of Figure 1 we back up our theoretical intuition and empirically demonstrate that autoencoders with categorical latent variables can achieve significantly more storage compression of input observations at the same average distortion as autoencoders with continuous variables. More detail is provided about this experiment in Appendix A.1.

## 2.3 Modeling Hippocampal Memory Index Theory

*Hippocampal memory index theory* was first proposed by Teyler & DiScenna (1986) as a theory for brain function in the hippocampus and later slightly revised in (Teyler & Rudy, 2007) after twenty years of related research. The theory primarily involves the hippocampus, believed to be involved in recalling previous experiences, and neocortex, believed to be responsible for reasoning. The crux of the theory is the idea that the hippocampus does not literally store previous experiences, but rather

efficiently stores light weight *indexes* corresponding to information that can then be retrieved from a complimentary *association cortex*. The *association cortex* is considered to be part of the human neocortex, however, its location varies by species. As a result, in our work we model it as a separate component of our system.

Our artificial neural network implementation of this model consists of three primary modules. For clarity, we provide an illustration of our proposed three module architecture in Figure 2. We model a reasoning module with a standard deep neural network that has an architecture suited to the current goal of the system. We also have an association module, which we model as a variational autoencoder (Kingma & Welling, 2014) with discrete latent variables as in (Jang et al., 2017; Maddison et al., 2017). The final component of our system is a recollection buffer that stores latent codes associated with prior experiences. In conjunction with the decoder of the association module, we can sample from the buffer to create approximate recollections. These recollections, for example, can be used to stabilize training for the reasoning and association modules. In the continual lifelong learning setting, we can keep a memory of the class index and task index associated with each input as they are already in a very compact representation. More generally, we will demonstrate using this capability to transfer old knowledge to a new machine learning model without fully storing any data.

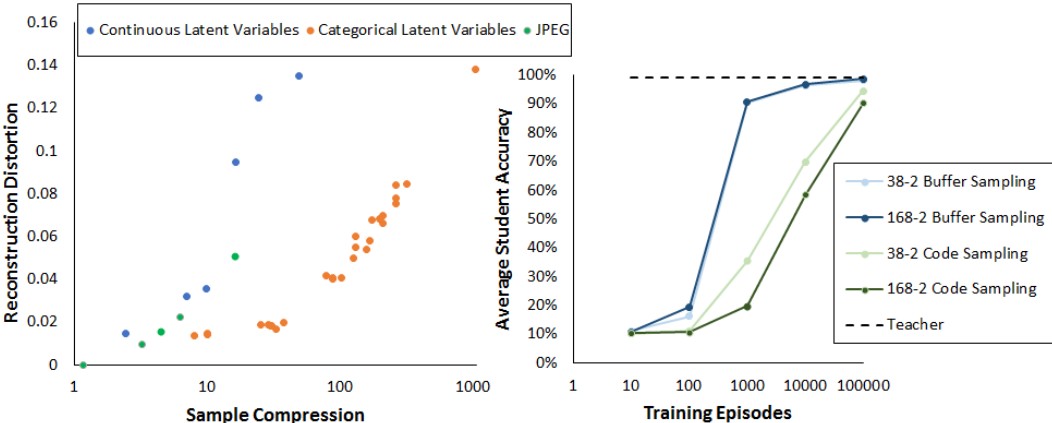

Figure 1: Left: A comparison of the relationship between average reconstruction L1 distance on the MNIST training set and sample compression for both continuous latent variable and categorical latent variable autoencoders. Right: Comparison of generative transfer learning performance using a CNN teacher and student model on MNIST while using code sampling and recollection buffer sampling.

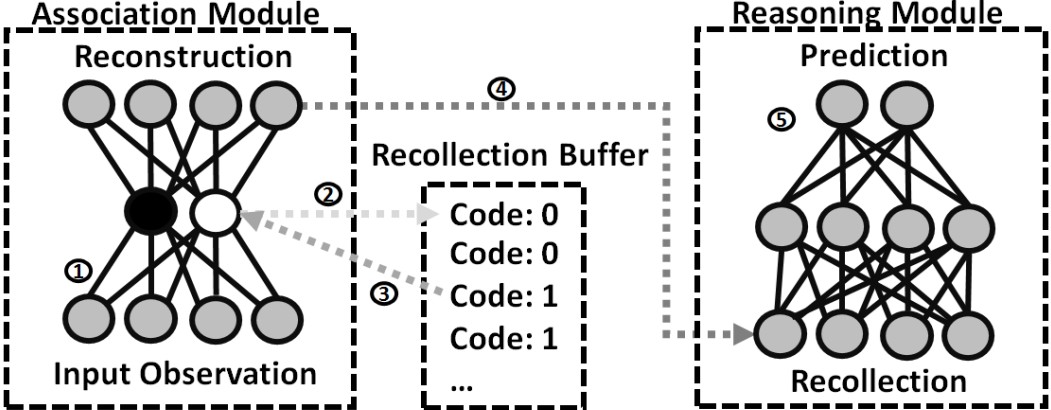

Figure 2: An example illustration of our proposed three module architecture. First, the input observation is encoded to a latent code that is stored in a recollection buffer. Later, a code is selected from the buffer and sent through the decoder to provide a recollection for which the reasoning module makes a prediction.

Obviously, a major hyperparameter of our proposed model is the latent code size of the autoencoder. In Appendix B we derive a maximization procedure that we use throughout our experiments to find the optimal autoencoder latent variable size to use for a given resource constraint. We also provide additional clarification about the setup for our knowledge distillation experiments in Figure 5 and our continual lifelong learning experiments in Figure 6.

## 2.4 SUITABILITY FOR DEEP LIFELONG LEARNING SYSTEMS

**Self generated recollections still stabilize the association model.** On the surface, it may seem reasonable to question whether recollections generated by the association model can actually be effective in preventing catastrophic forgetting for the very same model. However, this has been achieved in literature by leveraging the difference in model parameters across time scales of relevance to the problem. In fact, recently successful strategies for preventing catastrophic forgetting have relied on saving models specific to each prior task (Li & Hoiem, 2016; Rusu et al., 2016; Riemer et al., 2016; Kirkpatrick et al., 2017). In this work, we explore a strategy that is more generic and not reliant on human defined task boundaries to function correctly. Instead of just taking one gradient step per training example, we take $N$ steps on $N$ mini-batches drawn from the replay buffer. We sample all batches from the buffer before updating the association model, so that when $N > 1$ the parameters used to decode recollections are at a time delay with the model being trained. As $N$ grows large, the time delay becomes a significant regularization on the optimization of the model. As shown in the left chart of Figure 3 for continual learning on CIFAR-100 with $N = 10$ and an effective incremental buffer size of 200, self generated replay is very similarly effective to real replay for stabilizing the lifelong autoencoder. The negative effects of the less effective synthetic examples are apparently drowned out by the positive effects of a larger diversity of stored examples.

**Incremental storage needs decrease with more experiences.** In this paper, we conduct experiments over finite time horizons and thus benefit from some constant gain in efficiency for each problem to achieve superior results in comparison to full storage of experiences (which grows linearly with the number of experiences). When scaling to a truly massive number of experiences, we can tolerate a large startup cost if our scaling is less than linear with the number of experiences. As such, it is interesting to show not just the constant factor of greater efficiency between lossy storage and lossless storage at a given number of experiences, but also to show an increasing gap as the number of experiences grow. Because of correlation and thus transfer learning between past experiences and future experiences, we expect this to be the case in most situations where learning is possible. Transfer learning from past experiences leads to efficiency gains in that a small code size should be capable of closer approximation of the inputs after seeing more prior experiences. This in turn corresponds to fewer model parameters needed in the association model and fewer codes needed to represent each experience in the index buffer to maintain a reasonable degree of recollection distortion. We demonstrate this effect empirically in the right chart of Figure 3 for continual learning on CIFAR-100 using a small autoencoder model leveraging 76 2d categorical latent variables. In blue we show online training of the model with a random initialization and no buffer (online-

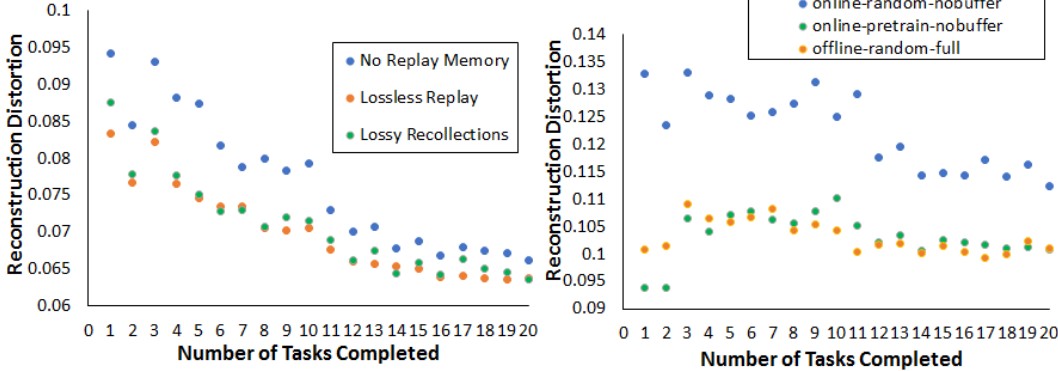

Figure 3: Left: Average test set L1 reconstruction distortion on incremental CIFAR-100 using an autoencoder with an effective incremental storage buffer size of 200. Right: Average test set L1 reconstruction distortion on incremental CIFAR-100 using an autoencoder with 76 2d categorical latent variables.

| Latent Representation | Sampling Strategy | Reconstruction Distortion | Nearest Neighbor Distortion |
|---|---|---|---|
| 38 2d variables | Code Sampling | 0.058 | 0.074 |
| | Buffer Sampling | 0.058 | 0.054 |
| 168 2d variables | Code Sampling | 0.021 | 0.081 |
| | Buffer Sampling | 0.021 | 0.021 |

Table 1: Comparing the nearest training example L1 distance of code sampling and buffer sampling based recollections. We report averages across 10,000 random samples. Reconstruction distortion of the autoencoder is measured on the test set and is not influenced by the sampling strategy.

random-nobuffer). In orange we show offline training with random initialization and full data storage (offline-random-full) trained over 100 iterations. Predictably, access to unlimited storage and all the tasks simultaneously means that the performance of offline-random-full is consistently better than online-random-nobuffer. To demonstrate the value of transfer from a good representation in the online setting, we show in green an online model with no replay buffer and a representation initialized after training for 100 iterations on CIFAR-10 (online-pretrain-nobuffer). We note that online-pretrain-nobuffer performs comparably to the best randomly initialized model with access to all tasks simultaneously and unlimited storage (offline-random-full). In fact, it performs considerably better for the first few tasks where the number of prior experiences is much greater than the number of new experiences. The improvements from transfer learning have a substantial effect on stabilizing a reasoning module as we achieve 0.7% improvement in retention accuracy over real replay with the online model and 6.7% improvement with the initialization from CIFAR-10.

# 3 ACHIEVING SAMPLE EFFICIENT KNOWLEDGE TRANSFER

## 3.1 NAVIGATING THE CAPACITY AND EXPLORATION TRADE-OFF

The central idea of *hippocampal memory index theory* has always been to model the human capabilities of pattern completion (i.e. auto-association) and pattern separation (i.e. the experience buffer). What is possibly less evident than the fact that humans have these combined capabilities is why the combination is useful for a deep lifelong learning system. In this section, we highlight a key utility in that it allows the association model to circumvent an inherent trade-off between latent variable capacity and exploration in achieving sample efficient knowledge transfer between networks.

The essence of the capacity and exploration trade-off is nicely highlighted for two categorical latent variable autoencoders on MNIST in Table 1. Let us first consider the typical method of sampling a variational autoencoder, which we will refer to as *code sampling*, where each latent variable is selected randomly. Obviously, by increasing the capacity of our autoencoder we are able to achieve lower reconstruction distortion. However, we interestingly find that items sampled from the autoencoder are even further from the closest item in the training distribution. It turns out that while increasing the autoencoder capacity increases modeling capabilities, it also increases the chance that a chosen latent code will not be representative of those seen in the training distribution. Keeping a buffer of indexes associated with prior experiences provides a light weight mechanism of keeping track of the statistical distribution of seen experiences while taking advantage of high dimensional causal features for modeling. In the right chart of Figure 1, we highlight the massive benefit of this capability in the sample efficiency of transferring a function with knowledge distillation. We would like to emphasize that these results are not a byproduct of increased model capacity associated with the buffer as the small model with the buffer significantly outperforms the big model with code sampling despite 7.4x less total bits of storage including the model parameters and buffer.

## 3.2 AUTOMATED GENERATIVE CURRICULUM LEARNING

While random sampling from a buffer can be very effective, we would like to further maximize the efficiency of distilling knowledge from a teacher model to a student model. This motivates the automated curriculum learning setting (Bengio et al., 2009) as recently explored for multi-task learning in (Graves et al., 2017) or rather automated generative curriculum learning in our case. We tried some simple reinforcement learning solutions with rewards based on (Graves et al., 2017) but were unsuccessful in our initial experiments because of the difficulty of navigating a complex continuous action space. We also tried an active learning formulation proposed for GANs to learn the best latent code to sample (Zhu & Bento, 2017) at a given time. We had limited success with this strategy as well as it tends to learn to emphasize regions of the latent space that optimize incorrectness, but no longer capture the distribution of inputs.

**Designing generative sampling heuristics.** Inspired by these findings, we instead employ simple sampling heuristics to try to design a curriculum with prototypical qualities like responsiveness to the student and depth of coverage. We model responsiveness to the student as *active sampling* by focusing on examples where the student does not have good performance. We randomly sample $k$ latent codes using our recollection buffer and choose the one that is most difficult for the current student for backpropagation by cheaply forward propagating through the student for each. By sampling from the recollection buffer, we are able to ensure our chosen difficult samples are still representative of the training distribution. We set $k$ to 10 in our experiments so the sampling roughly equates to sampling once from the most difficult class for the student model at each point in time. We model depth of coverage by sampling a bigger batch of random examples and adding a filtering step before considering difficulty. We would like to perform *diverse sampling* that promotes subset diversity when we filter from $kn$ examples down to $k$ examples. One approach to achieving this is a Determinantal Point Process (DPP) (Kulesza et al., 2012) as recently proposed for selecting diverse neural network mini-batches (Zhang et al., 2017). We use the dot product of the inputs as a measure of similarity between recollections and found the DPP to achieve effective performance as a diverse sampling step. However, we follow (Bouneffouf & Birol, 2015) and use a process for sampling based on the sum of the squared similarity matrix as outlined in Appendix D. We found the sum of the squared similarity matrix to be equally effective to the determinant and significantly more scalable to large matrices. We also set $n$ to 10 in our experiments.

## 4 RELATED WORK

**Generative Knowledge Distillation.** Even the first work on the topic of knowledge distillation (Bucilu et al., 2006) introduced a strategy for producing synthetic data to amplify real data. Additionally, unlabelled data has been widely used (Riemer et al., 2016; Laine & Aila, 2017; Ao et al., 2017; Kulkarni et al., 2017) for knowledge distillation. Generative models have also been used as a sole source for distillation before in the context of language models (Shin et al., 2017), but not in the more general case where there is a separate input and output to generate for each example. Graves et al. (2017) recently looked at the problem of automated curriculum learning in a multi-task setting. In this work, we model a setting where the curriculum designer is responsible for the constructing inputs and outputs as opposed to simply choosing among a set of them. In (Matiisen et al., 2017) they learn a task generator that is appropriate for curriculum learning, but do not learn a function that constructs synthetic inputs. By achieving high quality purely generative distillation, our goal is to obtain a form of general purpose knowledge transfer. As a result, our work is related in motivation to techniques that look to preserve knowledge after transforming the network architecture (Chen et al., 2015; Wei et al., 2016).

**Related Lifelong Models.** Pseudorehearsals (Robins, 1995) is a related approach for preventing catastrophic forgetting that does not require explicit storage of patterns. Instead it relies on learning a generative model alongside the main model. The generative model produces pseudo-experiences that are combined in batches with real experiences during training. As the true labels for pseudo-experiences are obviously not available, the current main model's representation is used to create a target not to forget. For simple learning problems, very crude approximation of the real data such as randomly generated data from an appropriate distribution can be sufficient. However, for complex problems like those found in NLP and Computer Vision with highly structured high dimensional inputs, more refined approximations are needed to stimulate the network with relevant old representations. The view taken in (Li & Hoiem, 2016) for Computer Vision and (Riemer et al., 2016) for NLP is that input generation can be a very challenging problem in its own right that can be side-stepped by using the data of the current task as inputs to prevent forgetting. As demonstrated in (Aljundi et al., 2016), this strategy works best when the inputs of the old task and new task are drawn from a similar subspace. Unfortunately, using the current data creates a large bias in the distribution used to prevent forgetting that cannot be suitable for truly non-stationary problems. In our work, we address these various concerns by proposing a novel pseudo-experience generator module that leverages episodic storage to efficiently model the statistics of more complex distributions of high dimensional observations. Recent work also looks at the problem of generative lifelong learning (Ramapuram et al., 2017) with a variational autoencoder, introducing a modified objective that would potentially be complementary to our contribution.

| Episodes | Real Data | 10% Sample | 2% Sample | 1% Sample | Real $x$ Teacher $y$ | 10x Compress | 50x Compress | 100x Compress |
|---|---|---|---|---|---|---|---|---|
| 10 | 10.43 | 9.94 | 11.07 | 10.70 | 10.07 | 10.65 | 10.99 | **13.89** |
| 100 | 19.63 | 18.16 | 22.82 | 22.35 | **25.32** | 19.34 | 16.20 | 21.06 |
| 1000 | 90.45 | 88.88 | 90.71 | 89.93 | **91.01** | 90.66 | 90.52 | 90.03 |
| 10000 | 97.11 | 96.83 | 95.98 | 94.97 | **97.42** | 96.77 | 96.37 | 95.65 |
| 100000 | 98.51 | 97.99 | 96.14 | 94.92 | **98.63** | 98.59 | 98.17 | 97.75 |

Table 2: Generative knowledge distillation random sampling experiments with a CNN teacher and student model on MNIST.

## 5 EMPIRICAL RESULTS

### 5.1 GENERAL PURPOSE KNOWLEDGE DISTILLATION EXPERIMENTS

We will now empirically demonstrate our proposed recollection generator's ability to transfer knowledge from a teacher neural network to a student model. In our experiments, we train a teacher model with a LeNet (LeCun et al., 1998) convolutional neural network (CNN) architecture on the popular MNIST benchmark, achieving 99.29% accuracy on the test set. Alongside the teacher model, we train a generator model with discrete latent variables. Each model is trained for 500 epochs. During the final pass through the data, we forward propogate through each training example and store the latent code in a recollection buffer. This buffer eventually grows to a size of 50,000. After training is complete, the recollection buffer is used as a statistical basis for sampling diverse recollections to train a student network. A logical and effective strategy for training a student model is to sample randomly from this buffer and thus capture the full distribution.

**Comparing the recollection buffer to lossless baselines.** In Table 2 we validate the effectiveness of our technique by comparing it to some lossless storage baselines of interest. As baselines we consider training with the the same number of randomly sampled real examples, using real input and the teacher's output vector as a target, and using random sampling to select a subset of lossless memories with equivalent storage footprints. When training with a large number of memories for a more complete knowledge transfer, the recollection compression clearly shows dividens over random sampling baselines. This is impressive particularly because these results are for the stricter total storage resource constraint setting and on a per sample basis the compression is actually 37x, 101x, and 165x to account for the autoencoder model capacity. We also would like to validate these findings in a more complex setting for which we consider distillation with outputs from a 50 task Resnet-18 teacher model that gets 94.86% accuracy on Omniglot. We test out performance after one million training episodes, which is enough to achieve teacher performance using all of the real training examples. However, sampling diversity restricts learning significantly, for example, achieving 28.87% accuracy with 10% sampling, 8.88% with 2% sampling, and 5.99% with 1% sampling. In contrast lossy compression is much more effective, achieving 87.86% accuracy for 10x total resource compression, 74.03% accuracy for 50x compression, and 51.45% for 100x compression.

**Distilling functions to different neural architectures.** Our main motivation for enabling general purpose knowledge distillation is for occasions where we would like to change the architectural form of our knowledge over time, not keep it constant. In Table 8 we consider distillation from our LeNet CNN teacher model to a multi-layer perceptron (MLP) student with two hidden layers of 300 hidden units. For MLPs we again see our recollection compression is comparable to the performance of real examples while using much less storage, and lossy compression scales much better than sampling lossless inputs.

**Active and Diverse Sampling.** In Table 3 we empirically demonstrate the superior performance of the proposed active and diverse sampling strategies discussed in Section 3.2. Consistently we are able to achieve more efficient training of the student networks than is achieved with random real examples. This is an interesting result implying that a reactive generative distillation teacher model that has mastered a skill can be more sample efficient in conveying knowledge to a student model than humans can be by labelling random unlabelled data.

### 5.2 CONTINUAL LIFELONG LEARNING WITHOUT FORGETTING EXPERIMENTS

To assess the ability of our recollection generator to stabilize continual lifelong learning without forgetting, we follow recent work (Lopez-Paz & Ranzato, 2017; Rebuffi et al., 2017) using the incremental CIFAR-100 dataset. It consists of 20 tasks made up of the course grained super class

| Episodes | Real Data | Real $x$ Teacher $y$ | Active 10x Compress | Active 100x Compress | Active & Diverse 10x Compress | Active & Diverse 100x Compress |
|---|---|---|---|---|---|---|
| 10 | 10.43 | 10.07 | 9.95 | 10.19 | 10.67 | **11.51** |
| 100 | 19.63 | 25.32 | 14.80 | 22.57 | 27.05 | **29.93** |
| 1000 | 90.45 | 91.01 | 93.45 | 92.97 | **94.81** | 92.54 |
| 10000 | 97.11 | 97.42 | **98.61** | 97.53 | 98.59 | 97.66 |
| 100000 | 98.51 | 98.63 | 99.18 | 98.25 | **99.20** | 98.32 |

Table 3: Generative knowledge distillation active and diverse sampling experiments with a CNN teacher and student model on MNIST. The real input baselines are randomly sampled.

| Model | Effective Buffer Size | Recollections | Retention |
|---|---|---|---|
| Online Resnet-18 | 0 | 0 | 33.3 |
| Resnet-18 Replay | 10 | 10 | 29.4 |
|  | 50 | 50 | 33.4 |
|  | 200 | 200 | 43.0 |
| Recollection Generator | 10 | 5000 | **39.7** |
|  | 50 | 5000 | **47.9** |
|  | 200 | 5000 | **51.6** |

Table 4: Lifelong learning retention results on incremental CIFAR-100 for low effective buffer sizes with an incremental storage resource constraint.

labels in CIFAR-100, which each contain 5 sub-classes. The tasks are learned in the predefined sequence incrementally one example at a time. In recent efforts on this dataset, the smallest number of examples stored in memory considered was 200, allowing memorization of on average two lossless examples from each class in the dataset. We will explore resource constrained settings like this and ideally even smaller incremental resource constrained settings to test out the efficacy of our recollection generator. We model our experiments after (Lopez-Paz & Ranzato, 2017) and use a Resnet-18 base model. All of our models store examples in their buffer following the reservoir sampling procedure.

**Lifelong learning with an incremental storage resource constraint.** In Table 4 we consider performance on incremental CIFAR-100 with a very small incremental resource constraint, including a couple of settings with even less incremental memory than the number of classes. A primary baseline of ours for comparison is replay which follows the same structure as our model, but with lossless recollections as opposed to lossy recollections. It performs relatively well when the number of examples is greater than the number of classes, but otherwise suffers from a biased sampling towards a subset of classes. This can be seen by the decreased performance for a buffer of size 10 from what is achieved with no buffer at all learning online. Consistently we see that tuning our recollection generator to lossy recollections with reasonably sized diversity results in improvements over real examples at the same incremental resource cost. To further validate our method against another dataset, we turned to Omniglot and learning in an incremental 50 task setting. Omniglot is more challenging for resource constrained continual lifelong learning than CIFAR-100 because it contains more tasks and fewer examples of each class. With an incremental resource constraint of 10 full examples, a Resnet-18 model with replay achieves 3.6% final retention accuracy (online learning produces 3.5% accuracy). In contrast, our recollection generator achieves 5.0% accuracy. For an incremental resource constraint of 50 full examples, replay achieves 4.3% accuracy which is further improved to 4.8% accuracy by taking three gradient descent steps per new example. The recollection generator once again achieves better performance with 9.3% accuracy at one step per example and 13.0% accuracy at three steps per example.

The value of using lossy recollections becomes even more apparent for long term retention of skills. We demonstrate this empirically in Figure 4 by first training models on incremental CIFAR-100 and then training them for 1 million training examples on CIFAR-10. In the end, the number of training examples seen from the original dataset is only 5% of the size of the examples seen from in the new domain. We find that larger lossy buffers retain knowledge more effectively than smaller lossless buffers when switching to the new domain. For example, the 1200 example replay buffer looses knowledge faster than the 200 effective buffer size recollection generator despite better performance when originally training on incremental CIFAR-100. CIFAR-10 is a challenging transfer task for preventing catastrophic forgetting as catastrophic forgetting is most severe for experiences that have large dot products in the representation space with new experiences (French, 1992).

| Model | Recollections | Retention |
|---|---|---|
| Resnet-18 Replay | 200 | 43.0 |
| GEM (Lopez-Paz & Ranzato, 2017) | 200 | 48.7 |
| iCaRL (Rebuffi et al., 2017) | 200 | 43.6 |
| 76 2d Categorical Variables - No Transfer | 1392 | 43.7 |
| 76 2d Categorical Variables - CIFAR-10 Transfer | 1392 | **49.7** |

Table 5: Lifelong learning retention results on incremental CIFAR-100 with a 200 lossless episode total storage resource constraint. GEM and iCaRL baselines are from (Lopez-Paz & Ranzato, 2017).

**Lifelong learning with a total storage resource constraint.** Given our results so far, it may be fair to criticize our introduction of autoencoder parameters and question whether these incremental savings will truly be realized in practice. On the other hand, to demonstrate performance with a very small total resource constraint on a single dataset, an incredibly small autoencoder would then be needed to learn from scratch a function for the very complex input space. We demonstrate in Table 5 that transfer learning provides a solution to this problem. By transferring in background knowledge we are able to perform much better at the onset with a small autoencoder. We explore the total resource constraint of 200 examples that is the smallest explored in (Lopez-Paz & Ranzato, 2017) and demonstrate that we are able to achieve state of the art results by initializing only the autoencoder representation with one learned on CIFAR-10. CIFAR-10 is drawn from the same larger database as CIFAR-100, but is non-overlapping. We should emphasize that the technique proposed in our paper is some sense orthogonal to and possibly complimentary to approaches to utilizing these episodic examples explored in (Rebuffi et al., 2017; Lopez-Paz & Ranzato, 2017). However, we provide them for comparison to highlight that the gains seen by leveraging efficient lossy recollections for this problem outweigh those related to making more sophisticated use of the recollections.

## 6 CONCLUSION

We have proposed a discrete latent variable autoencoder based model for generative knowledge transfer and stabilizing lifelong learning without forgetting. We demonstrated that autoencoders with discrete latent variables are capable of far more sample compression than continuous latent variable models. We also highlighted that variational autoencoders equipped with a lossy recollection buffer are capable of multiple orders of magnitude faster knowledge distillation in comparison to traditional code sampling. We consistently find lossy compression of memories to be more effective than lossless episodic sampling. However, they are also complimentary in a resource constrained lifelong learning setting, and we get best results in the human realistic setting of having both lossy and sampled memories. For lifelong learning it is important to efficiently scale resources for learning over a very large number of examples. Our initial very promising results using transfer learning demonstrate that the incremental memory footprint for new memories can decrease for a lifelong learning agent over time. This is a key ingredient to scaling deep lifelong learning systems and is a promising direction for future studies to expand upon.

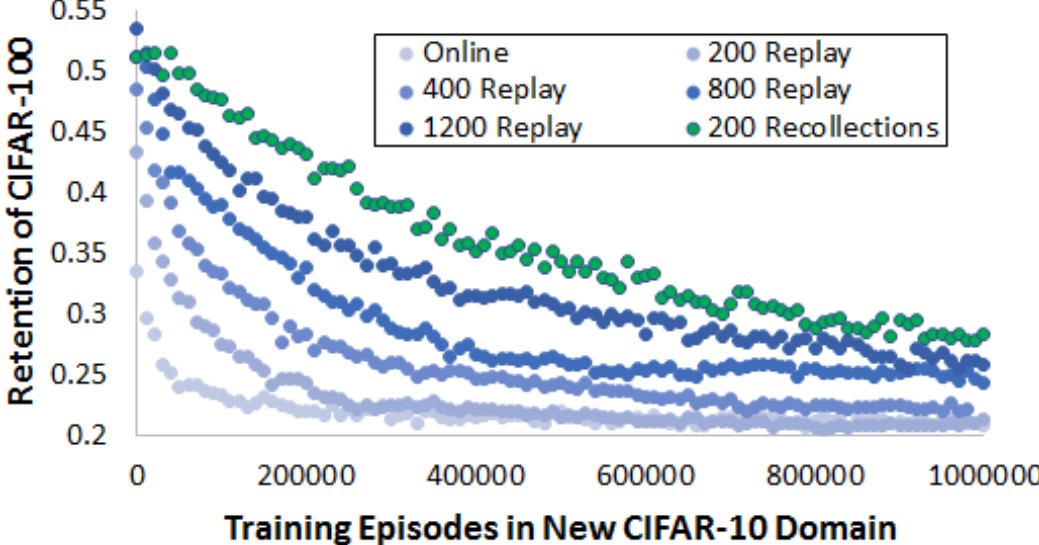

Figure 4: Retention of performance on CIFAR-100 after prolonged training on CIFAR-10. We compare lossy and lossless replay buffer strategies listed by their effective incremental buffer size.

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

| $c$ | $l$ | Compression | Distortion |
|---|---|---|---|
| 6 | 20 | 209.067 | 0.06609 |
| 10 | 20 | 125.440 | 0.04965 |
| 6 | 16 | 261.333 | 0.07546 |
| 12 | 10 | 130.667 | 0.05497 |
| 10 | 14 | 156.800 | 0.05410 |
| 24 | 3 | 130.667 | 0.05988 |
| 38 | 2 | 165.053 | 0.05785 |
| 6 | 2 | 1045.333 | 0.13831 |
| 40 | 3 | 78.400 | 0.04158 |
| 20 | 2 | 313.600 | 0.08446 |
| 8 | 6 | 261.333 | 0.08423 |
| 12 | 6 | 174.222 | 0.06756 |
| 30 | 2 | 209.067 | 0.06958 |
| 24 | 6 | 87.111 | 0.04065 |
| 4 | 37 | 261.333 | 0.07795 |
| 8 | 15 | 196.000 | 0.06812 |
| 48 | 10 | 32.667 | 0.01649 |
| 209 | 8 | 10.003 | 0.01455 |
| 12 | 37 | 87.111 | 0.03996 |
| 313 | 4 | 10.019 | 0.01420 |
| 392 | 3 | 8.000 | 0.01348 |
| 50 | 18 | 25.088 | 0.01859 |
| 168 | 2 | 37.333 | 0.01955 |
| 108 | 3 | 29.037 | 0.01894 |
| 62 | 2 | 101.161 | 0.04073 |
| 208 | 2 | 30.154 | 0.01832 |
| 68 | 5 | 30.745 | 0.01849 |

Table 6: This table provide more specifics about the discrete latent variable architectures involved in Figure 1.

## A    ADDITIONAL DETAILS ON EXPERIMENTAL PROTOCOL

Each convolutional layer has a kernel size of 5. As we vary the size of our categorical latent variable across experiments, we in turn model the number of filters in each convolutional layer to keep the number of hidden variables consistent at all intermediate layers of the network. In practice, this implies that the number of filters in each layer is equal to $cl/4$. We note that the discrete autoencoder is stochastic, not deterministic and we just report one stochastic pass through the data for each experimental trial.

### A.1    DISTORTION AS A FUNCTION OF COMPRESSION EXPERIMENTS

More detail about the architecture used in these experiments are provided for categorical latent variables in Table 6 and for continuous latent variables in Table 7. For each architecture we ran with a learning rate of 1e-2, 1e-3, 1e-4, and 1e-5, reporting the option that achieves the best training distortion. For the distortion, the pixels are normalized by dividing by 255.0 and we take the mean over the vector of the absolute value of the reconstruction to real sample difference and then report the mean over the samples in the training set. Compression is the ratio between the size of an 8bpp MNIST image and the size of the latent variables, assuming 32 bits floating point numbers in the continuous case and the binary representation as in (1) for the categorical variables. The JPEG data points were collected using the Pillow Python package using quality 1, 25, 50, 75, and 100. We subtracted the header size form the JPEG size so it is a relatively fair accounting of the compression for a large data set of images all of the same size. The JPEG compression is computed as an average over the first 10,000 MNIST training images.

| $h$ | Compression | Distortion |
|----|----|----|
| 1 | 49 | 0.135196 |
| 2 | 24.5 | 0.124725 |
| 3 | 16.33333333 | 0.0947032 |
| 5 | 9.8 | 0.0354035 |
| 7 | 7 | 0.031808 |
| 20 | 2.45 | 0.0149272 |

Table 7: This table provide more specifics about the continuous latent variable architectures involved in Figure 1.

## A.2 MNIST GENERATIVE DISTILLATION EXPERIMENTS

For all of our distillation experiments we ran the setting with a learning rate of 1e-3 and 1e-4, reporting the best result. We found that the higher learning rate was beneficial in setting with a low number of examples and the lower learning rate was beneficial in setting with a larger number of examples. The categorical latent variable autoencoders explored had the following representation sizes: 168 2d variables for 10x compression, 62 2d variables for 50x compression, and 38 2d variables for 100x compression. For our code sampling baselines, we used the numpy random integer function to generate each discrete latent variable.

## A.3 OMNIGLOT GENERATIVE DISTILLATION EXPERIMENT

The learning rate for the Resnet-18 reasoning model was 1e-4 in our experiments. Our trained discrete autoencoder models were of the following representation sizes: 32 variables of size 2 for 100x compression, 50 variables of size 2 for 50x compression, and 134 variables of size 2 for 10x compression. We follow 90% multi-task training and 10% testing splits for Omniglot established in (Yang & Hospedales, 2017).

## A.4 INCREMENTAL RESOURCE CONSTRAINT EXPERIMENTS

Our categorical latent variable autoencoders had the following sizes for incremental CIFAR-100: 48 2d variables for an effective buffer size of 10, 244 2d variables for an effective buffer size of 50, and 620 3d variables for an effective buffer size of 200. The reasoning model was trained with a learning rate of 1e-3 in all of our experiments. The learning rate for the autoencoder was 1e-4 for the buffer size of 200 and 1e-3 for other buffer sizes.

For incremental Omniglot our learning rate was set to 1e-3. For the effective buffer size of 50 experiments, we leveraged a categorical latent variable autoencoder with 312 2d variables. For the effective buffer size of 10 experiments, we utilized a categorical latent variables consisting of 62 2d variables. We follow 90% multi-task training and 10% testing splits for Omniglot established in (Yang & Hospedales, 2017).

## A.5 INCREMENTAL CIFAR-100 TOTAL RESOURCE CONSTRAINT EXPERIMENTS

During the transfer learning experiments from CIFAR-10, a learning rate of 1e-3 was used for the Resnet-18 reasoning model and a learning rate of 3e-4 was used for the discrete autoencoder generator. For the experiment without transfer learning, we instead used a higher learning rate of 1e-3 for the autoencoder.

## B OPTIMIZING LATENT CODE SIZE FOR A RESOURCE CONSTRAINT

The ability of a discrete variational autoencoder to memorize inputs should be strongly related to the effective bottleneck capacity $C_{\mathrm{ve}}$, which we define, for discrete latent variables, as:

$$C_{\mathrm{ve}} = \log_2 l^c. \tag{4}$$

### B.1    Incremental Storage Resource Constraints

First, let us consider the dynamics of balancing resources in a simple setting where we have an incremental storage constraint for new incoming data without regard for the size of the model used to compress and decompress recollections. We refer to the total storage constraint over all $N$ incoming examples as $\gamma$ and the average storage rate limit as $\gamma/N$. We can then define $\rho$ as the probability that an incoming example is stored in memory. Thus, the expected number of bits required per example stored is $\rho S_{\mathrm{sbe}}$, assuming simple binary encoding. If we treat $\rho$ as fixed, we can then define the following optimization procedure to search for a combination of $c$ and $l$ that maximizes capacity while fulfilling an incremental resource storage constraint:

$$
\begin{aligned}
\underset{c,l}{\text{maximize}} \quad & C_{\mathrm{ve}} \\
\text{subject to} \quad & \rho S_{\mathrm{sbe}} \leq \frac{\gamma}{N},
\end{aligned}
\tag{5}
$$

which yields the approximate solution $C_{\mathrm{ve}} \simeq \frac{\gamma}{N\rho}$. As seen in equation 5, there is an inherent trade-off between between the diversity of experiences we store governed by $\rho$ and the distortion achieved that is related to the capacity. The optimal trade-off is likely problem dependent. Our work takes a first step at trying to understand this relationship. For example, we demonstrate that deep neural networks can see improved stabilization in resource constrained settings by allowing for some degree of distortion. This is because of an increased ability to capture the diversity in the data at the same incremental resource constraint.

### B.2    Total Storage Resource Constraints

In some ways, the incremental storage constraint setting described in the previous section is not the most rigorous setting when comparing lossy compression to lossless compression where a subset of full inputs are selected. Another important factor is the number of parameters in the model $|\theta|$ used for compression and decompression. $|\theta|$ generally is also to some degree a function of $c$ and $l$. For example, in most of our experiments, we use the same number of hidden units $cl$ at each layer as used in the bottleneck layer. With fully connected layers, this yields $|\theta|(c,l) \propto (cl)^2$. As such, we can revise equation 5 to handle a more rigorous constraint for optimizing a discrete latent variable autoencoder architecture:

$$
\begin{aligned}
\underset{c,l}{\text{maximize}} \quad & C_{\mathrm{ve}} \\
\text{subject to} \quad & \rho S_{\mathrm{sbe}} + |\theta|(c,l) \leq \gamma/N,
\end{aligned}
\tag{6}
$$

While this setting is more rigorous when comparing to lossless inputs, it is a somewhat harsh restriction with which to measure lifelong learning systems. This is because it is assumed that the compression model's parameters should be largely transferable across tasks. To some degree, these parameters can be viewed as a sunk cost from the standpoint of continual learning. In our experiments, we also look at transferring these representations from related tasks to build a greater understanding of this trade-off.

## C    Additional Details About The Buffer

To provide additional details about the setup of our experiments, we have included Figure 5, illustrating our architecture in the *knowledge distillation* setting, and Figure 6, explaining how we combine examples for a batch update in the *continual lifelong learning* setting.

## D    Minimum Sum of Squared Similarities

This algorithm is trying to find a new landmark point that maximizes the determinant by finding a point that minimizes the sum of squared similarities (MSSS). The MSSS algorithm initially randomly chooses two points from the dataset $X$. It then computes the sum of similarities between

the sampled points and a subset, $T$, selected randomly from the remaining data points. The point with the smallest sum of squared similarities is then picked as the next landmark data point. The procedure is repeated until a total of $m$ landmark points are picked.

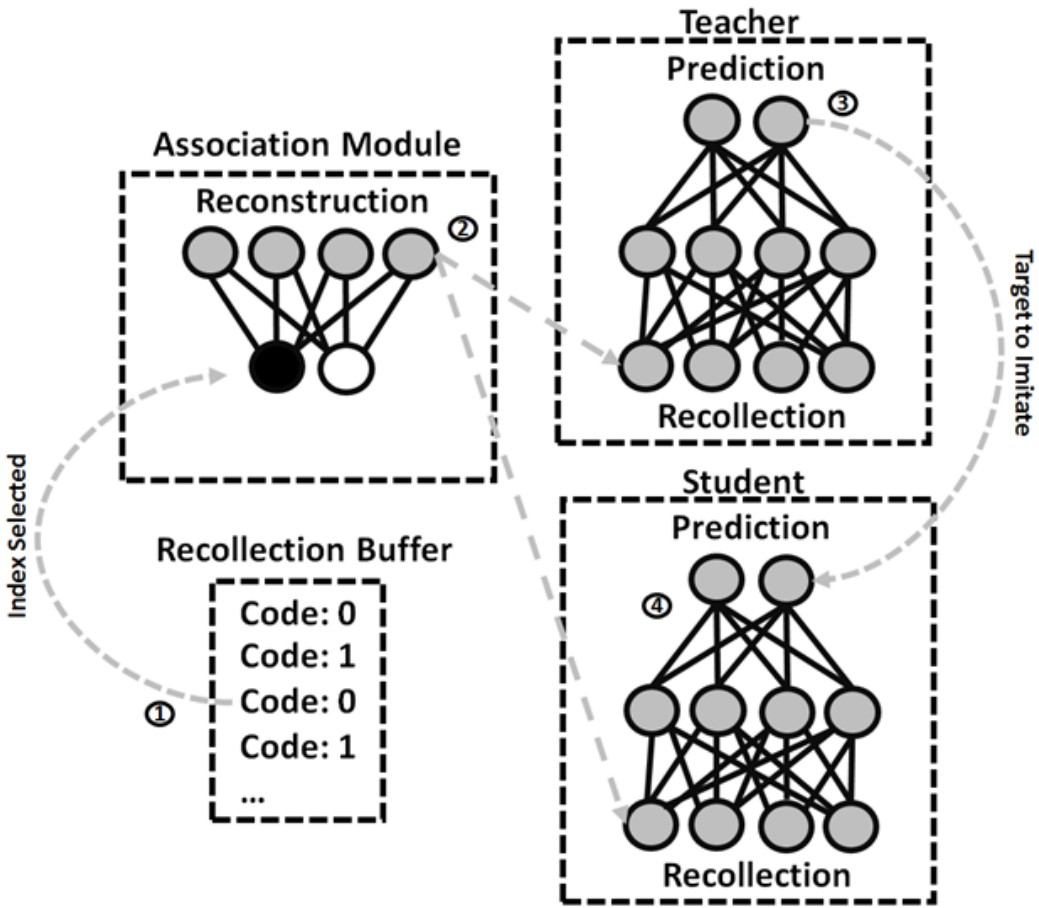

Figure 5: An illustration of how our proposed recollection generator is used to produce recollections to transfer knowledge from a teacher neural network to a student neural network.

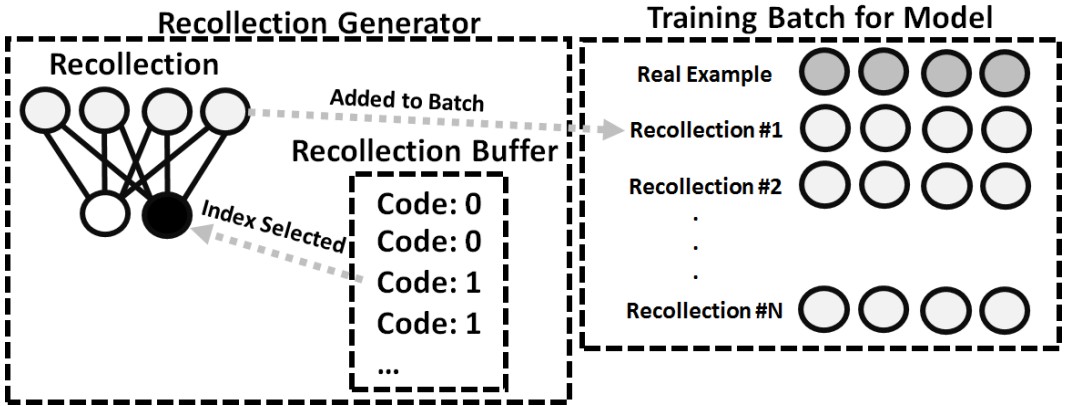

Figure 6: An illustration of how our proposed recollection generator is used to produce recollections that are interleaved with real incoming examples for training continual lifelong learning models.

| Episodes | Real Data | 10% Sample | 2% Sample | 1% Sample | Real $x$ Teacher $y$ | 10x Compress | 50x Compress | 100x Compress |
|---|---|---|---|---|---|---|---|---|
| 10 | 13.64 | **17.04** | 14.57 | 15.13 | 15.87 | 16.70 | 11.80 | 14.66 |
| 100 | 36.37 | 37.04 | 38.35 | 34.04 | 38.56 | 37.16 | 40.09 | **42.31** |
| 1000 | 80.54 | 79.08 | 78.18 | 77.76 | 80.00 | **80.72** | 80.00 | 77.75 |
| 10000 | 91.04 | 90.84 | 88.38 | 86.83 | 90.86 | **91.37** | 90.60 | 90.46 |
| 100000 | 96.66 | 95.02 | 91.61 | 88.97 | 96.60 | **96.71** | 96.24 | 95.22 |

Table 8: Generative knowledge distillation random sampling experiments with a CNN teacher and MLP student model on MNIST.

---

**Algorithm 1** The Minimum Sum of Squared Similarities Algorithm

---

1: **Input:** $X = \{x_1, x_2, ..., x_n\}$: dataset
2: $m$: number of landmark data points
3: $\gamma$: size of the subsampled set from the remaining data, in percentage
4:
5: **Output:** $\widetilde{S} \in R^{m \times m}$: similarity matrix between landmark points
6: Initialize $\widetilde{S} = I_0$
7: **For (i=0 to i<2) do**
8: $\quad \widetilde{x}_i = Random(X)$
9: $\quad \widetilde{S} := \widetilde{S}_{\cup x_i}$
10: $\quad \widetilde{X} := \widetilde{X} \cup \{\widetilde{x}_i\}$
11: **End For**
12: **While** $i < m$ **do**
13: $\quad T = Random(X \setminus \{\widetilde{X}\}, \gamma)$
14: $\quad$ Find $\widetilde{x}_i = argmin_{x \in T} \sum_{j < i-1} sim^2(x, \widetilde{x}_j)$
15: $\quad \widetilde{S} := \widetilde{S}_{\cup \widetilde{x}_i}$
16: $\quad \widetilde{X} := \widetilde{X} \cup \{\widetilde{x}_i\}$
17: **End While**

---

# E  CNN TO MLP DISTILLATION RESULTS

In Figure 8 we explore generative knowledge distillation transferring knowledge from a CNN teacher network to a MLP student network.

