# OpenReview forum: "Generation and Consolidation of Recollections for Efficient Deep Lifelong Learning"
_ICLR.cc/2018/Conference — Reject_

### Official Review · AnonReviewer1 · 2017-11-27
**Recollections for efficient deep lifelong learning**

**Rating:** 5
**Confidence:** 2

**Review:**

The paper proposes an architecture for efficient deep lifelong learning. The key idea is to use recollection generator (autoencoder) to remember the previously processed data in a compact representation. Then when training a reasoning model, recollections generated from the recollection generator are used with real-world examples as input data. Using the recollection, it can avoid forgetting previous data. In the experiments, it has been shown that the proposed approach is efficient for transfer knowledge with small data compared to random sampling approach.

It is an interesting idea to remember previous examples using the compact representation from autoencoder and use it for transfer learning. However, I think the paper would be improved if the following points are clarified.

1. It seems that reconstructed data from autoencoder does not contain target values. It is not clear to me how the reasoning model can use the reconstructed data (recollections) for supervised learning tasks.

2. It seems that the proposed framework can be better presented as a method for data compression for deep learning. Ideally, for lifelong learning, the reasoning model should not forget previously learned kwnoledge embeded in their weights.
However, under the current architecture, it seems that the reasoning model does not have such mechanisms.

3. For lifelong learning, it would be interesting to test if the same reasoning model can deal with increasing number of tasks from different datasets using the recollection mechanisms.

---

> ### Author Response · Authors · 2018-01-05
> **Response to Comments By Reviewer 1**
>
> We have attempted to address your concern about retention of knowledge by the reasoning model when it is presented with many additional experiences. In Figure 4, we plot CIFAR-100 model performance after switching from continual lifelong learning on CIFAR-100 to the disjoint set of labels from CIFAR-10 for many training examples. Our results highlight that the increased diversity of experiences helps the resource constrained system retain knowledge better when using lossy storage than it does when using comparable lossless storage techniques.
>
> In our continual lifelong learning experiments, we store the task and label index along with the latent code in the recollection buffer, as this information is already very light weight. We have reformatted the presentation of the approach to make this clearer in the paper.
>
> Regarding the benefit of the reasoning model not forgetting previously learned knowledge, we would first comment that our approach makes very few assumptions about the reasoning model. This feature would likely be orthogonal and complimentary to our approach in many settings. However, we would like to highlight that our goal isn’t only to prevent forgetting. Our goal is to navigate the stability-plasticity dilemma in a way that maximizes performance on old and new examples. Experience replay provides an approximation of i.i.d. stationary random input sampling in non-stationary environments, allowing neural networks to effectively optimize for the true objective with the efficacy of offline training in the limit of an unbounded experience buffer size. (Lopez-Paz & Ranzato, NIPS 2017) found EwC (Kirkpatrick et al., PNAS 2017) a popular forgetting prevention technique to be ineffective relative to techniques leveraging episodic storage for continual lifelong learning on CIFAR-100. One of the big reasons they found for the performance difference was that EwC focuses on retaining its poor performance on early tasks, while techniques with episodic storage continually improve on old examples as they learn relevant concepts later.

---

### Official Review · AnonReviewer2 · 2017-11-29
**This paper presents important and timely problem of lifelong learning under resource constraints; the manuscript lacks clarity and structure; limited novelty.**

**Rating:** 5
**Confidence:** 3

**Review:**

This paper addresses lifelong learning setting under resource constraints, i.e. how to efficiently manage the storage and how to generalise well with a relatively small diversity of prior experiences. The authors investigate how to avoid storing a lot of original training data points while avoiding catastrophic forgetting at the same time.
The authors propose a complex neural network architecture that has several components. One of the components is a variational autoencoder with discrete latent variables, where the recently proposed Gumbel-softmax distribution is used to efficiently draw samples from a categorical distribution (Jang et al ICLR 2017). Discrete variables are categorical latent variables using 1-hot encoding of the class variables. In fact, in the manuscript, the authors describe one-hot encoding of c classes as l-dimensional representation. Why is it not c-dimentional? Also the class probabilities p_i are not defined in (7).
This design choice is reasonable, as autoencoder with categorical latent variables can achieve more storage compression of input observations in comparison with autoencoders with continuos variables.
Another component of the proposed model is a recollection buffer/generator, a generative module (alongside the main model) which produces pseudo-experiences. These self generated pseudo experiences are sampled from the buffer and are combined with available real samples during training to avoid catastrophic forgetting of prior experiences. This module is inspired by episodic training proposed by Lopez-Paz and Ranzato in ICLR2017 for continual learning. In fact, a recollection buffer for MNIST benchmark has 50K codes to store. How fast would it grow with more tasks/training data? Is it suitable for lifelong learning?

My main concern with this paper is that it is not easy to grasp the gist of it. The paper is 11 pages long and often has sections with weakly related motivations described in details (essentially it would be good to cut the first 6 pages into half and concentrate on the relevant aspects only). It is easy to get lost in unimportant details, where as important details on model components are not very clear and not structured. Second concern is limited novelty (from what I understood).

---

> ### Author Response · Authors · 2018-01-05
> **Response to Comments By Reviewer 2**
>
> Thank you for your comment about the beginning of the paper. We have significantly reorganized the way we present the ideas to address your comment. Hopefully this also helps highlight some of the novel ideas presented in this paper. Our approach is novel in that it is the first that models hippocampal memory index theory using modern deep neural networks. We are also the first to demonstrate how the theory’s signature combination of pattern completion and pattern separation work together to enable faster knowledge transfer using recollections. This capability, in turn, leads to a model that can efficiently distill its knowledge to a student network of a different architecture without storing any real examples. Additionally, it can enable more effective experience replay with superior scaling in resource constrained settings of continual lifelong learning.
>
> We have also tried to address your confusion related to the description of the Gumbel-Softmax function. To clarify, we are using c variables that are each l dimensional, implying we use c separate one hot encodings of size l to represent a latent code. This is standard practice for discrete latent variable autoencoders. We adopt conventions from (Jang et al., ICLR 2017) where possible in our presentation of the approach. We have also reworked the presentation of our approach to make the scaling considerations clear. A key benefit of the proposed technique we argue for in section 2.4 is that because of transfer learning, scaling is less than linear with the number of experiences in contrast with the linear scaling of storing lossless experiences.

---

### Official Review · AnonReviewer4 · 2017-12-07
**Deep Lifelong learning with recollections under resource constraints.**

**Rating:** 5
**Confidence:** 3

**Review:**

This paper presents an approach to lifelong learning with episodic experience storage under resource constraints. The key idea of the approach is to store the latent code obtained from a categorical Variational Autoencoder as opposed to the input example itself. When a new task is learnt, catastrophic forgetting is avoided by randomly sampling stored codes corresponding to past experience and adding the corresponding reconstruction to a batch of data from a new problem. The authors show that explicitly storing data provides better results than random sampling from the generative model. Furthermore, the method is compared to other techniques relying on episodic memory and as expected, achieves better results given a fixed effective buffer size due to being able to store more experience.

While the core idea of this paper is reasonable, it provides little insight into how episodic experience storage compares to related methods as an approach to lifelong learning. While the authors compare their method to other techniques based on experience replay, I feel that a comparison to other techniques is important. A natural choice would be a model which introduces task-specific parameters for each problem (e.g.  (Li & Hoiem, 2016) or (Rusu et al., 2016)).

A major concern is the fact that the VAE with categorical latents itself suffers from catastrophic forgetting. While the authors propose to "freeze decoder parameters right before each incoming experience and train multiple gradient descent iterations over randomly selected recollection batches before moving on to the next experience", this makes the approach both less straight-forward to apply and more computationally expensive.

Moreover, the authors only evaluate the approach on simple image recognition tasks (MNIST, CIFAR-100, Omniglot). I feel that an experiment in Reinforcement Learning  (e.g. as proposed in (Rusu et al., 2016)) would provide more insight into how the approach behaves in more challenging settings. In particular, it is not clear whether experience replay may lead to negative transfer when subsequent tasks are more diverse.

Finally, the manuscript lacks clarity. As another reviewer noted, detailed sections of weakly related motivations fail to strengthen the reader's understanding. As a minor point, the manuscript contains several grammar and spelling mistakes.

---

> ### Author Response · Authors · 2018-01-05
> **Response to Comments By Reviewer 4**
>
> We appreciate your concern that the VAE itself would suffer from catastrophic forgetting. We have attempted to provide more clarification about how the VAE is able to stabilize itself with self-generated recollections. We have also provided two additional charts to serve as empirical evidence that this happens when training in a continual lifelong learning setting on CIFAR-100. In the left chart of Figure 3, we demonstrate that self-generated recollections can stabilize a lifelong autoencoder as well as real example replay of a comparable resource footprint, and significantly better than online training on CIFAR-100. In Figure 4, we demonstrate that after running our CIFAR-100 models for many training examples on CIFAR-10, the benefit of the extra diversity of experiences we can have when using lossy recollections to prevent forgetting outweighs negative effects associated with forgetting that the VAE experiences.
>
> Thank you for your question about freezing the decoder parameters before each incoming experience. We have now made this clearer in section 2.4. Instead of freezing the decoder parameters and keeping two copies, we can simply forward propagate for all of the replay mini-batches associated with learning the current example ahead of time. This feature, as well as the encoding and decoding of memories and training of the autoencoder, do indeed add computation over lossless methods. In our experiments, however, these costs were pretty negligible as the computation associated with the larger Resnet-18 reasoning model overshadows the computation associated with our much smaller VAEs. While this strategy does indeed add some computation for our approach, it is also critical for stabilizing the training of the autoencoder for continual lifelong learning, as we explain in section 2.4.
>
> The techniques you mentioned for comparison are, unfortunately, not suited for the resource constrained lifelong learning problems we explore in this paper. We tried LwF for continual learning saving the old model parameters after every task on CIFAR-100 and found it to be very ineffective in terms of performance. It is also very computationally expensive later in training as the number of terms in the loss function grow linearly with the number of tasks. This result aligns with the experiments in (Lopez-Paz & Ranzato, NIPS 2017) that found a similar forgetting prevention technique EwC (Kirkpatrick et al., PNAS 2017) to be less effective than episodic techniques for lifelong learning on CIFAR-100. This is largely because forgetting prevention techniques focus on retaining poor performance on early tasks while episodic storage techniques continue to improve on these tasks when they learn relevant concepts later.
>
> Progressive Neural Networks have not been shown to scale to the number of tasks and deep residual network architectures that we consider. This is because model parameters scale even more than linearly with the number of tasks due to lateral connections with all prior task representations at each layer. As a result, each new task adds more parameters than the task before it. Our approach is not reliant on human defined tasks to work. Additionally, incremental storage and computation costs from adding model parameters with each task for such a large model consumes far more resources than the episodic storage footprints we consider in our experiments. We should also note that the work of (Rusu et al., 2016) is not directly comparable to ours in that their few task reinforcement learning experiments are not performing continual learning. They use A3C, which may be superior to experience replay methods in terms of wall clock time for convergence. However, A3C involves multiple agents performing RL episodes at the same time on different threads, which is not the same as continual learning of a single agent. While A3C is fast in terms of wall clock time, it is not efficient with the total number of episodes needed to reach good performance. On the other hand, this kind of efficiency is an important criteria for the very difficult and ambitious task of continual lifelong learning.
>
> While this work does not address the related topic of alleviating negative transfer in multi-task learning, our work does provide a clear advancement in the study of experience replay mechanisms for lifelong learning. Experience storage has been a key component to stabilize training of many of the most successful lifelong learning and reinforcement learning algorithms to date. It is not the goal of this paper to compare this very successful family of methods with other alternatives that function quite differently.

---

### Author Response · Authors · 2018-01-05
**New Revisions to Address Reviewer Concerns**

We would like to thank the reviewers for their time and feedback. To address reviewer concerns about clarity, we substantially reorganized and edited the paper. We hope our revised draft makes both the novelty and motivation of our approach clearer. There are substantial differences with the earlier version due to an adjusted presentation structure, but we did not significantly change the ideas presented.

We will now directly address the concerns raised by each reviewer.

---

### Decision · Program_Chairs · 2018-01-29
**ICLR 2018 Conference Acceptance Decision**

**Decision:**

Reject

**Comment:**

The reviewers were uniformly unimpressed with the contributions of this paper. The method is somewhat derivative and the paper is quite long and lacks clarity. Moreover, the tactic of storing autoencoder variables rather than full samples is clearly an improvement, but it still does not allow the method to scale to a truly lifelong learning setting.